# Bi-level Graphs for Cellular Pattern Discovery

**Zhenzhen. Wang**[*]

**Aleksander S. Popel**[†]

**Jeremias. Sulam**[‡]

## Abstract

The tumor microenvironment is widely recognized for its central role in influencing cancer progression and prognostic outcomes. Despite extensive research efforts dedicated to characterizing this complex and heterogeneous environment, considerable challenges persist. In this study, we introduce a novel data-driven approach for identifying tumor microenvironment patterns that are closely tied to patient prognoses. Our methodology relies on the construction of a bi-level graph model to integrate information across different scales: (i) a cellular graph, which models the intricate tumor microenvironments, and (ii) a population graph that captures inter-patient similarities, given their respective cellular graphs, by means of a soft Weisfeiler-Lehman kernel. We demonstrate our approach in breast cancer patients, obtain data-driven risk stratification, and identify crucial patterns associated with patient prognosis. This method provides valuable insights into the prognostic implications of the breast tumor microenvironment and holds the potential to analyze other cancers.

## 1 Introduction

The tumor microenvironment (TME) is a complex ecosystem, comprising proliferating tumor cells, tumor stroma, immune cells, blood vessels, and lymphatics [1]. There is accumulating evidence underscoring the pivotal role of the TME in driving tumor progression [2], contributing to treatment resistance [3], and influencing patient prognosis [4]. Recent technological advancements in spatial multiplex proteomics, such as imaging mass cytometry (IMC) [5], have enabled the simultaneous assessment of a wide spectrum of proteins in tissue specimens, which allow for a comprehensive exploration of the complexity and heterogeneity of TMEs at the single-cell level [6].

Increasing attention has been given to studying the TME at varying scales [7, 8], including the analysis of cell type compositions in various cancers [9, 10, 11], the quantification of spatial distances

---

[*]Department of Biomedical Engineering
Johns Hopkins University
Baltimore, MD 21210
zwang218@jhu.edu
[†]Department of Biomedical Engineering
Johns Hopkins University
Baltimore, MD 21210
apopel@jhu.edu
[‡]Department of Biomedical Engineering
Johns Hopkins University
Baltimore, MD 21210
jsulam1@jhu.edu

NeurIPS 2023 AI for Science Workshop.

between pairs of cell phenotypes [12], and the characterization of cellular neighborhoods involving more than two cell types [13, 14, 15]. While these works have unveiled numerous unique patterns across a spectrum of cancers, associating them with clinical implications still relies on the formulation of explicit hypotheses regarding the relationship between the TME and diseases, grounded in domain expertise and prior knowledge [16, 9, 17], which may naturally and inadvertently constrain the exploration of novel relationships and patterns.

On the other hand, there is a growing interest in harnessing graph-based deep learning techniques to analyze the association between the TME and disease without the need for explicit hypotheses in a data-driven manner [18]. Several studies have reported promising results by levering graph neural networks (GNNs) to model the TME and predicting patients' clinical subtypes, outcomes, and survival [19, 20, 21, 22]. Yet, the high cost of multiplexed imaging technologies often limits the size of the patient cohort, which can severely constrain the quality of modern machine learning models by limiting their generalization, particularly in cross-studies scenarios [8]. Additionally, many of these studies employ post-hoc interpretation methods to obtain clinical implications from their models, focusing on the extraction of information related to the learned relationships [18, 19, 20]. Consequently, achieving meaningful explanations and interpretations for these results, which are contingent on accurate and well-calibrated predictions, becomes exceptionally challenging.

This study circumvents some of these limitations above by proposing a novel data-driven and unsupervised learning approach to stratify patients and unveil TME patterns relevant to prognosis. We refer to this method as **BiGraph**, which entails the construction of two interconnected graphs: a patient-specific *cellular graph*, and a subsequent *population graph* given the patients' characteristics captured by the former. The cellular graph meticulously models the TME for individual patients, capturing detailed information about the spatial proximity and phenotypic characteristics of cells. On the other hand, the population graph captures similarities among all patients given their TME patterns, with strong connections indicating high similarities. A novel graph kernel function, referred to as the *Soft-WL kernel*, serves as the bridge between these two hierarchies of graphs, measuring the similarity between pairs of cellular graphs. The combination of these two levels of graphs facilitates the identification of patient subgroups with similar TME patterns through community detection methods. In turn, the distinct survival outcomes observed among different patient subgroups provide valuable insights into the underlying associations between TME patterns and patients' prognoses.

## 2 Methods

Bigraph takes as input spatial coordinates and phenotypes of cells, and outputs a risk stratification of patients as well as the prognosis-relevant patterns that characterize each subgroup. The primary innovation of BiGraph revolves around exploiting relations across levels of graphs, namely, a cellular graph and a population graph, and their integration by means of a graph kernel method, as we outline in the section.

### 2.1 Preliminaries

We first present some key notation and definitions. A graph $G = (V, E)$ is defined by a tuple of nodes $V$ and a set of edges $E \subseteq \{\{u, v\} \subseteq V \mid u \neq v\}$. The set of nodes and edges in $G$ are denoted as $V(G)$ and $E(G)$, respectively. For each node $v$, the set of nodes with an edge connected to a node $v$ is defined as its neighborhood, denoted as $N(v)$.

The structure of a graph can be fully characterized by its *adjacency matrix*, denoted as $A$. An adjacency matrix with only binary entries represents a *binary* graph. Conversely, an adjacency matrix with continuous scalar entries represents a *weighted* graph, where $A_{uv}$ indicates the weight of the edge connecting nodes $u$ and $v$.

The graph $G$ can be decomposed into many *subgraphs*, defined as $G' := (V', E')$, where $V' \subseteq V$ and $E' \subseteq E$. The *subtree* is a special form of subgraph with a tree structure, which typically includes a root node $v$, and all other nodes included in the subtree are connected to $v$.

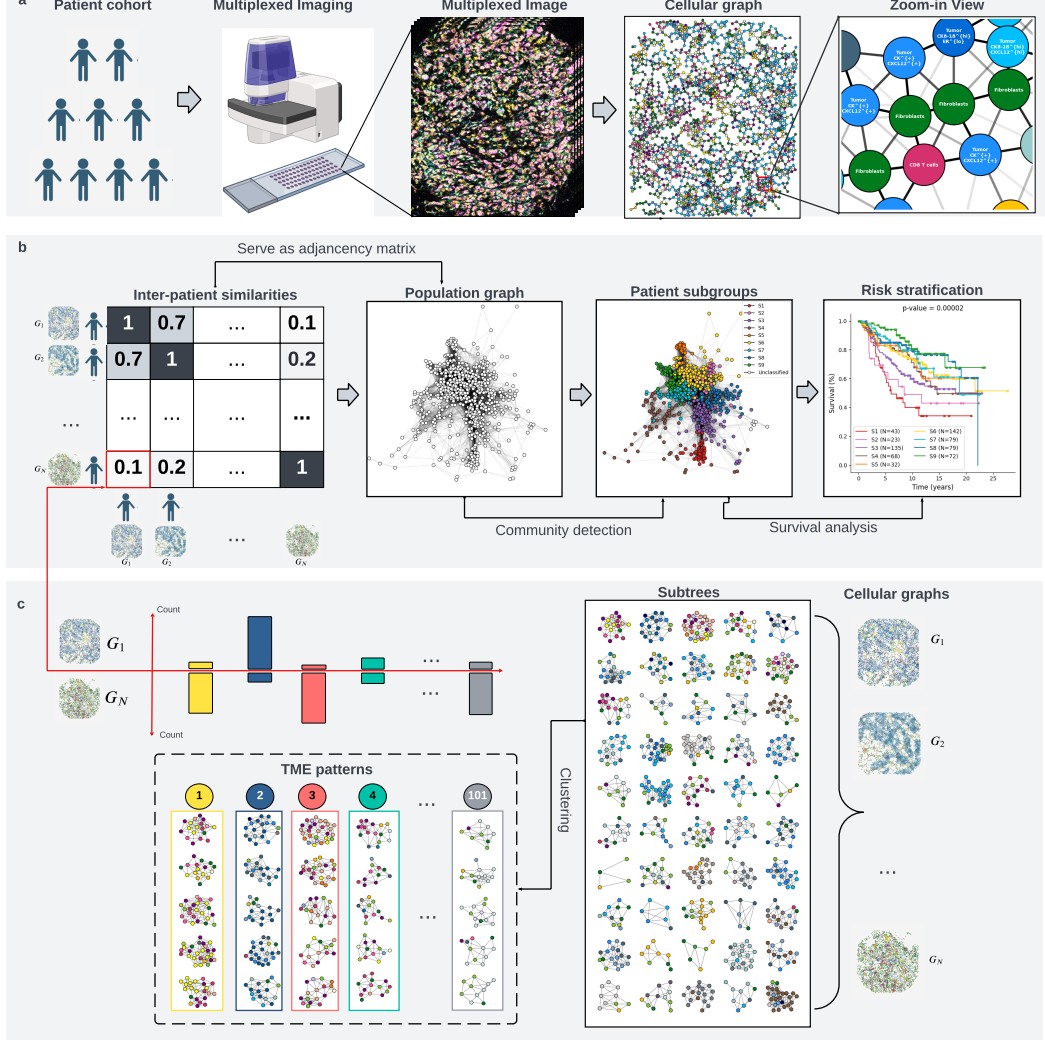

Figure 1: **Bi-level graphs bridged by the graph kernel (a)** The cellular graph is constructed to model patients' tumor microenvironment (TME) based on their multiplexed images. Each node represents a cell, and cells are connected via edges with varying weights inversely correlated with inter-cellular distance. **(b)** The population graph takes the inter-patient similarity matrix as its adjacency matrix, where each node is a patient, and edge weight represents the inter-patient similarity. Community detection methods are applied to the population graph to identify patient subgroups. Subsequent survival analysis provides risk stratification. **(c)** The Soft-WL subtree kernel method measures inter-patient similarities by comparing the TME pattern histograms of two patients. Cellular graphs are decomposed into subtrees. Similar subtrees are clustered to form a TME pattern, characterized by unique cell type composition and structure.

## 2.2 Construction of the cellular graph

The cellular graph models the TME of each individual patient, with each node in it corresponding to a cell and characterized by its spatial coordinates and phenotype label. Nodes (i.e., cells) are connected through edges, representing inter-cellular interactions (See Figure 1.a). Unlike conventional approaches that employ fixed distance thresholds to determine cell connectivity, we construct a complete cellular graph where all possible pairs of cells are potentially connected, with the strength of interaction decreasing as the distance between cells increases. To be more specific, the weight of

the edge connecting two cells $u$ and $v$, denoted as $w_{uv}$, is calculated via a Gaussian kernel given by

$$w_{uv} = \exp(-a\|d\|_2^2), \tag{1}$$

where $d$ represents the spatial distance between cells $u$ and $v$ in micrometers ($\mu$m), and $a$ is a parameter that controls how fast these weights decrease with distance, which we set to 0.01 in experiments.

## 2.3 Soft-WL subtree kernel: measuring inter-patient similarity

As described above, every patient is now represented by a cellular graph, and a novel graph kernel method called Soft-WL subtree kernel is used to measure the inter-patient similarities. The Soft-WL subtree kernel is a relaxation of the well-known Weisfeiler-Lehman (WL) kernel [23], designed to handle weighted cellular graphs and provide a smoother comparison between subtrees, as we detail in the following.

The Soft-WL subtree starts with a neighboorhood aggregation process to update the node features iteratively. Consider a cellular graph $G = (V, E)$ with a weighted adjacency matrix $A$. Noticeably, self-loops are added to each node, and thus the diagonal entries of $A$ are all 1. Each node, $v$, is initially assigned the one-hot encoding of its cell phenotype, denoted as $x_v^{(0)} \in \mathbb{R}^d$. The node feature is then iteratively updated in a series of iterations as follows:

$$x_v^{(h)} = x_v^{(h-1)} + \sum_{u \in V \setminus \{v\}} x_u^{(h-1)} A_{uv}, \tag{2}$$

where $h$ and $h - 1$ are the indices of iteration.

Intuitively, the neighborhood aggregation process enables the updated node feature to not only encode the phenotype information of itself but also its neighbors. Thus, the updated node feature, $x_v^{(h)}$, can also be viewed as the feature embedding of a subtree rooted at $v$. The boundary of this subtree is established by assessing the impact of neighborhood nodes $u$ on the embedding $x_v^{(h)}$. Equation (2) can be rewritten as

$$x_v^{(h)} = x_v^{(0)}(A^h)_{vv} + \sum_{u \in V \setminus \{v\}} x_u^{(0)}(A^h)_{uv}, \tag{3}$$

where $A^h$ denotes the $h$th power of the adjacency matrix $A$. Thus, the impact of a node $u$ to the subtree embedding $x_v^{(h)}$ is $A_{uv}^h$. A user-defined threshold $w_0$ (e.g., 0.1 in our experiments) is used to decide the boundary, where any node $u$ with $(A^h)_{uv} > w_0$ is considered a leave of the subtree.

Unlike the WL kernel, which only counts on isomorphic subtrees as a similarity measure, the Soft-WL subtree kernel clusters similar subtrees based on their feature embeddings. Every resultant cluster is considered a *TME pattern* containing subtrees with similar embeddings (see Figure 1.c). Furthermore, each TME pattern is assigned a "signature" given by averaging the embeddings of all the intra-cluster subtrees. The signature is a characterization of the cell phenotype composition and spatial organization of its corresponding TME pattern. The Soft-WL Subtree kernel calculates the inter-patient similarity by comparing the occurrence of these TME patterns across patients, which is formally defined as,

$$k(G_1, G_2) = \frac{\langle c(G_1, \Sigma), c(G_2, \Sigma) \rangle}{\|c(G_1, \Sigma)\|_2 \|c(G_2, \Sigma)\|_2}, \tag{4}$$

where $k(G_1, G_2)$ denotes the similarity between $G_1$ and $G_2$ (i.e., the similarity between the two corresponding patients), $\Sigma$ is the set of TME pattern identities, $c(G_1, \Sigma)$ and $c(G_2, \Sigma)$ denotes the histogram of TME patterns in $G_1$ and $G_2$, respectively, and $\|c(G_1, \Sigma)\|_2$ and $\|c(G_1, \Sigma)\|_2$ denote their L2 norms. The denominator serves as a normalization such that the similarity score for any patient pair falls within the range of 0 to 1, with two identical graphs achieving a maximum similarity score of 1.

The iteration $h$ is a hyperparameter that controls the size of subtrees, and we set it as $h = 2$ in the experiments. Although the clustering method can be general, we particularly use the PhenoGraph algorithm [24] to conduct subtree clustering since it does not need the number of clusters to be specified.

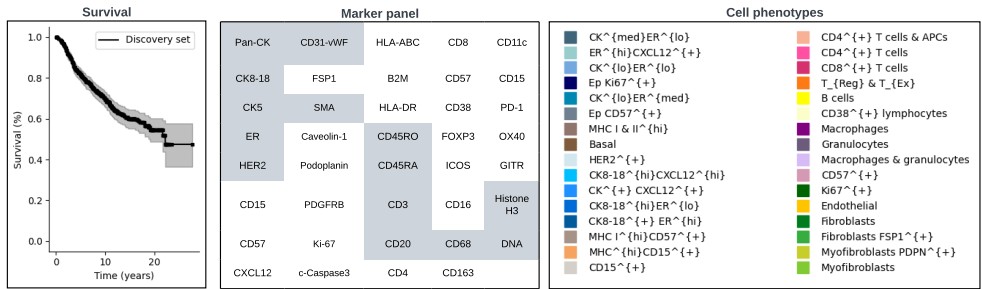

Figure 2: **Survival, marker panel, and phenotyping system of the dataset**

## 2.4 Population graph and patient subgroups

The population graph is constructed to model inter-patient similarities in the entire patient population (see Figure 1.b). Each node of the population graph represents a patient, and every patient is connected to potentially all other patients through edges with varying weights. Each edge weight represents the similarity between the two patients measured by the Soft-WL subtree method, given by Equation (4).

The Louvain community detection method [25] is employed to detect communities from the population graph, where each community detected represents a patient subgroup characterized by high intra-group similarities in their TMEs. The population graph is preprocessed by removing any edges with a weight lower than 0.5 to increase sparsity. The resolution of the Louvain algorithm is set to be 1 following common practice. The result of this process is the detection of patient subgroups, each of which represents a subset of patients with high intra-group similarities in their TMEs.

## 2.5 Discovery of prognosis-relevant patterns

Survival analysis is applied to each individual patient subgroup to assess its relative risk, and a risk stratification of the patient cohort can be obtained. A highlight of this methodology is its explainability: since the TME pattern histogram that is used to characterize each patient is completely transparent. Therefore, revisiting the TME pattern distribution in patient subgroups with distinct survivals – either better or worse – might unveil the underlying association between TME patterns and prognosis. To be more specific, we calculate the relative presentation of each TME pattern in a specific patient subgroup, which is defined as the ratio of its average occurrence among intra-group patients to that among all the patients. TME patterns with a relative presentation higher than 2 (i.e., twice) are considered "over-presented" in that patient subgroup. Furthermore, over-presented TME patterns in patient subgroups with distinct survivals are considered prognosis-relevant patterns and undergo further characterization and analysis.

# 3 Results

Although the methodology is general, we center our study in the context of breast cancer, using a patient cohort curated by E. Danenberg et al. [13]. It encompasses 693 breast cancer patients, each accompanied by their 37-dimensional imaging mass cytometry (IMC) images of tissue microarray cores and clinical data. This publicly available dataset provides cell phenotyping results, delineating 32 major cell phenotypes. Figure 2 provides an overview of the clinical information, marker panel, and phenotyping systems of this dataset.

## 3.1 Population graph analysis provides data-driven risk stratification

As elucidated in Section 2.2 and Section 2.3, a cellular graph is constructed for each patient to model the spatial distribution of different types of cells within the TME. the Soft-WL subtree kernel decomposes cellular graphs into numerous subtrees. Subtrees are clustered based on their feature embeddings calculated via neighborhood aggregation, culminating in a total of 102 clusters. Each

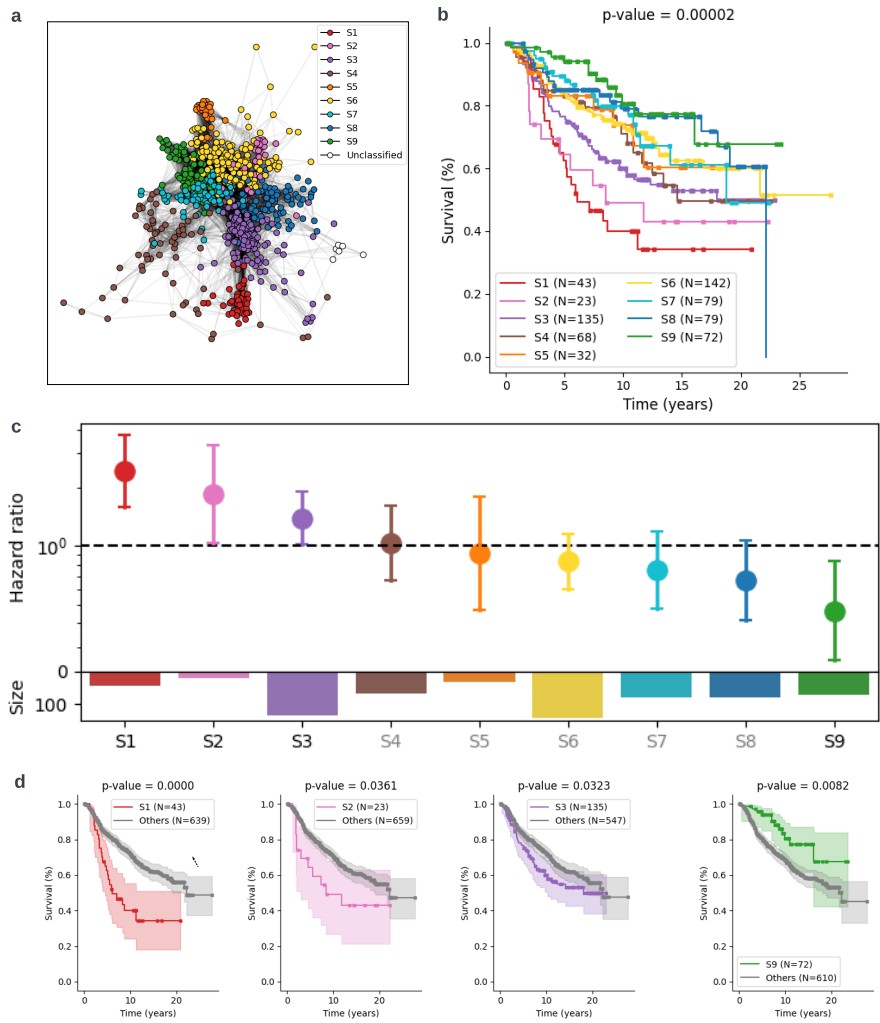

Figure 3: **Population graph provides risk stratification.** (a) 2-D visualization of the population graph and the results of community detection. **b** Survival plots of the 9 patient subgroups are displayed. A multivariate log-rank test is used to compare the 9 curves, and the resultant p-value is indicated in the title. **c** Relative hazard ratio (with 95% confidence interval) of the 9 patient subgroups estimated using a Cox proportional hazard model are shown. The text color (grey vs. black) distinguishes between statistically significant and non-significant associations (black: statistically significant). The size (i.e., number of patients) of each subgroup is presented in the corresponding barplot. **d** Survival plots (with 95% confidence interval) of 3 subgroups with significantly worse survival (S1, S2, and S3), and 1 subgroup with significantly better survival (S9) are shown.

of these clusters is considered a TME pattern, distinguished by a unique cell type composition and inter-cellular connections. A *signature* is assigned to each TME pattern by computing the average embedding of all corresponding subtrees, which summarizes the composition and spatial organization of different types of cells in subtrees with a specific pattern. The Soft-WL subtree kernel quantifies inter-patient similarity by comparing the occurrence of these 102 patterns in their respective TMEs (See Equation 4).

Having measured the similarities between any possible pair of patients in the discovery set, we construct a population graph that encodes the inter-patient similarities across the entire discovery set. In this graph, each node represents an individual patient, which is connected to other nodes via edges with varying weights corresponding to their similarity. To identify patient subgroups characterized by similar TME patterns, we employ the popular Louvain community detection method [25] on

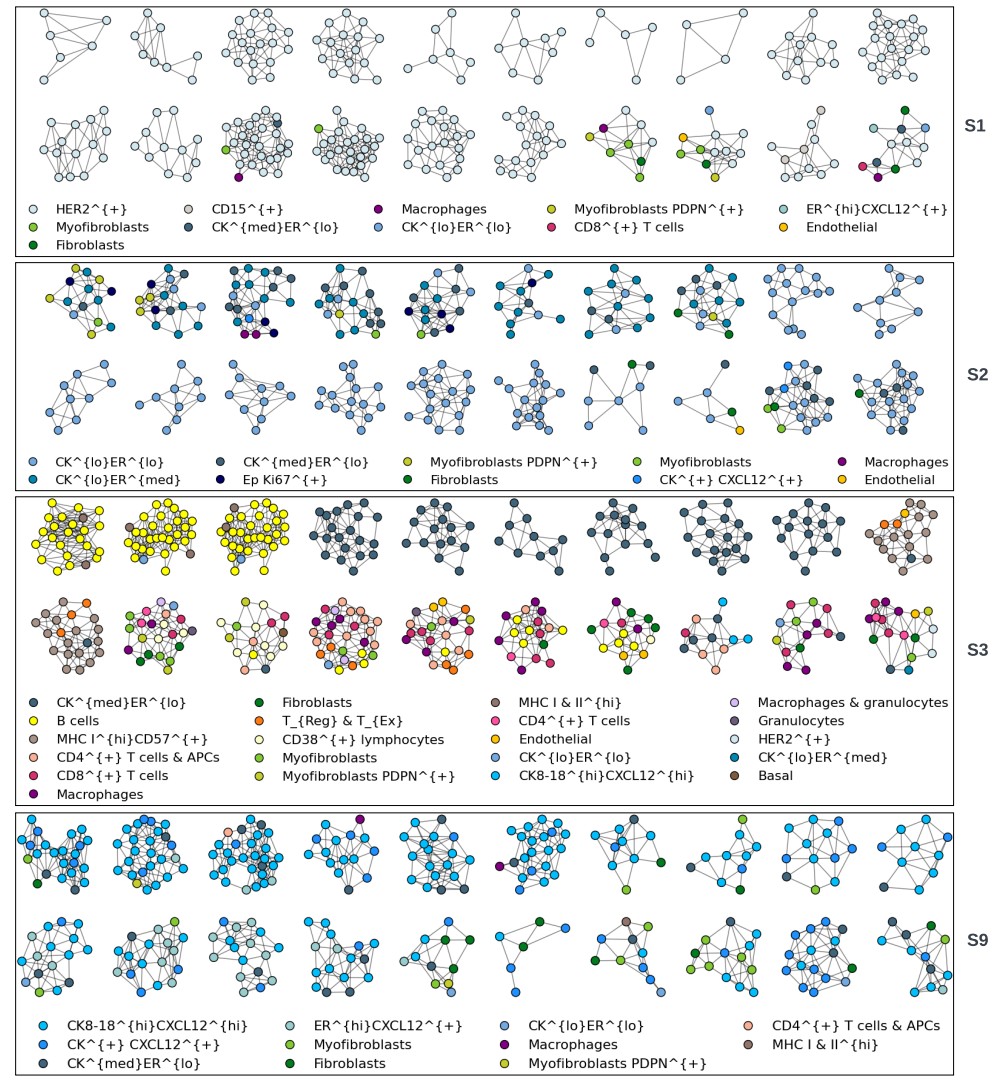

Figure 4: **Visualization of prognosis-relevant patterns.** This figure presents representative subtree examples of the over-presented TME patterns in four patient subgroups: three with significantly worse survival (S1, S2, and S3) and one subgroup with significantly better survival (S9).

the population graph. This process yields nine detected communities with different sizes, each representing a subgroup of patients with high intra-group similarities. For visualization purposes, we use the Fruchterman-Reingold force-directed algorithm [26] to assign virtual coordinates to nodes. A visualization of the population graph and the results of community detection are depicted in Figure 3.a.

The Hazard Cox model [27] is used to assess the prognostic significance of these patient subgroups. We rank the nine subgroups based on the estimated hazard ratio and designate them as S1 to S9 (Figure 3.c). Survival plots for these nine subgroups are generated using the Kaplan-Meier estimator [28], and we conduct a multivariate log-rank test [29] to compare their survival outcomes. Our results indicate a statistically significant difference in survival outcomes among these subgroups (Figure 3.b). Furthermore, we perform pairwise log-rank tests to compare the survival of patients within and outside each specific subgroup. These results reveal that S1, S2, and S3 exhibit significantly worse survival outcomes, and S9 has significantly better survival (Figure 3.d).

### 3.2 BiGraph unveils prognosis-relevant patterns

To unveil the association between TME patterns and survival outcomes, the relative presentation of TME patterns within each patient subgroup is analyzed, defined as the ratio of the pattern's mean occurrence per patient within that subgroup to the mean occurrence per patient across the entire cohort. We define TME patterns with a relative presentation higher than 2 as "over-presented" in the patient subgroup. Over-presented TME patterns in the four patient subgroups with statistically distinct survival outcomes (i.e., S1, S2, S3, and S9) are considered associated with prognosis. Some representative subtree examples corresponding to these prognosis-relevant TME patterns are shown in Figure 4. Notably, S1, which exhibits the worst survival among all subgroups, displays over-presented Her2$^+$ tumor cell clusters with varying sizes and cell densities. S2, associated with significantly worse survival, exhibits several over-presented patterns, most of which involve the CK$^{low}$ER$^{low}$ tumor cells. These patterns encompass homotypic CK$^{low}$ER$^{low}$ tumor cell clusters, heterotypic cell clusters comprising CK$^{low}$ER$^{med}$ tumor cells, Ki67$^+$ tumor cells, and PDPN$^+$ myofibroblasts, as well as co-occurrence of CK$^{low}$ER$^{med}$, CK$^{med}$ER$^{low}$, and CK$^{low}$ER$^{low}$ cells). S3, which exhibits slightly worse survival compared to the entire cohort, demonstrates a diverse range of over-presented TME patterns, including clusters of CK$^{med}$ER$^{low}$ tumor cells, B cell clusters, and immune hotspots featuring macrophages, CD4$^+$ T cells, CD8$^+$ T cells, and antigen-presenting cells (APCs). S9, characterized by the best survival outcome, demonstrates representative patterns characterized by the co-occurrence of CK8/18$^{high}$CXCL12$^{high}$ cells and CK$^+$CXCL12$^+$ cells. These over-presented TME patterns within patient subgroups exhibiting distinct survival outcomes are considered prognosis-relevant patterns.

## 4 Discussion and Conclusions

The TME represents a fundamental component of the study of cancer biology, and a number of recent studies indicate its promise for discovering biomarkers for treatment resistance and prognosis [30, 3, 4, 16, 31, 17]. While some useful metrics have been derived from pathology by employing traditional methods of spatial statistics, driven mostly by hypotheses relying on domain expertise, the discovery of data-driven markers for prognosis has remained unexplored. Modern deep learning methods, including graph neural networks, constitute an appealing alternative for data-driven biomarkers for prognosis. Yet, they typically require very large amounts of training data, can be sensitive to different experimental settings with low generalization, and provide limited information about the input data that cause a given predicted response.

This study develops a data-driven methodology, centered on breast cancer, that relies on two simple observations: *i)* representative patterns of the TME can be adaptively learned by studying the underlying patient-specific cellular graph, and *ii)* the relative presence of such patterns in different patients can be employed to provide a measure of similarity among these, from which a population-level graph can be constructed. This bi-level process allows us to obtain – automatically and in an unsupervised manner – different patient subgroups, with similar TMEs and potentially distinct prognoses, either better or worse. In turn, the presentation of TME patterns that characterize each of these subgroups unveils the underlying associations between TME and cancer prognosis. A significant strength of our methodology, compared with other data-driven methods, is the complete transparency of the features that provide the risk stratification and characterize better and worse survival. These features are dubbed as *TME patterns*, and they represent similar subtrees – cellular neighborhoods with a tree structure – identified in the TMEs. As a result, we can easily unveil the most (and least) common structures that characterize better (and worse) survival. Relying on more complex models, such as those based on graph neural networks, would have made it significantly different – if not altogether impossible – to easily characterize such biomarkers.

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
