# OpenReview forum: "Bi-level Graphs for Cellular Pattern Discovery"
_NeurIPS.cc/2023/Workshop/AI4Science — NeurIPS2023-AI4Science Poster_

### Official Review · Reviewer_yTwC · 2023-10-22
**Novel method that is not heavily data dependant; More results are needed**

**Rating:** 7
**Confidence:** 3

**Review:**

The authors propose a bi-level graph model called BiGraph that uses image mass cytometry (IMC) to analyze patterns in the Tumor microenvironment (TME) that lead to risk-based stratification of patients. This model is applied to 693 breast cancer patients to identify 9 patient subgroups that are diverse in terms of survival.

Major comments:
1. The proposed model is novel and can theoretically work in settings with limited data, which is a major challenge in cancer survival analysis.
2. While the authors have shown novel subtypes in the breast cancer cohort and associated them with respective marker cells, it would also be interesting to see how much of the information is lost when compared to GNNs in settings that have sufficient data. That being said, the proposed model is indeed easy to analyze and draw meaningful conclusions from for rare cancers.

Minor comments:
1. The problem is well-motivated and BiGraph is well-explained.

---

### Official Review · Reviewer_Z6zq · 2023-10-24
**Review of Submission101**

**Rating:** 5
**Confidence:** 3

**Review:**

__Summary__: The authors propose a novel graph vectorization method based on distance-weighted message passing, subtree clustering, and subtree motif counting within the graph. The authors apply this vectorization to create a graph similarity metric, and further apply this metric toward patient subtype identification on breast cancer slides. The resulting graph-based subtypes are prognostic and can be interpreted in terms of over-represented subgraphs, seeming to represent to prominent tissue components from the sample.

__Strengths__:
- The method is elegant and sample efficient. It is well justified by biological insights about the importance of the tumor microenvironment, and makes novel contributions by drawing upon well established literature.
- The results are impactful, relating tissue-level analysis to patient prognosis in breast cancer.
- The method provides an intuitive approach for vectorizing graphs, allowing evaluation of sufficiently large sets of graphs in terms of widely used distance and similarity metrics. This appears to be an important contribution, but its application is only explored for tumor subtyping with breast cancer slides, a very limited task for this general method.

__Weaknesses__:
- The method makes no comparison against existing subtyping methods (e.g. TCGA) or simpler methods like counting the abundance of certain cell types. The subtypes are certainly prognostic, but it is unclear if this is because of the proposed method or if simpler methods could reproduce this result, such as simply clustering on cell type proportions or using prognostic biomarkers such as HER2+ and EGFR expression among others.
- The authors seem to use subgraphs and subtrees interchangeably. This makes sense for $h=1$, but for $h=2$ how are the subgraphs guaranteed to be subtrees?

__Nits__:
- How does the method process graphs with different numbers of cells/nodes and different total subtrees?
- Figure 2 is confusing and has a somewhat vague caption.

__Recommendation__: Although the method is elegant and novel prognostic subtypes relating to tissue organization would be impactful, there are no competing works discussed. While the graph-based subtypes are prognostic, it is hard to tell if they improve prognosis. I recommend the authors add baseline comparisons or new applications of the method with comparisons before accepting this work.